# Relapsing High—Grade Glioma from Peritumoral Zone: Critical Review of Radiotherapy Treatment Options

**DOI:** 10.3390/brainsci12040416

**Published:** 2022-03-22

**Authors:** Maria Chiara Lo Greco, Roberto Milazzotto, Rocco Luca Emanuele Liardo, Grazia Acquaviva, Madalina La Rocca, Roberto Altieri, Francesco Certo, Giuseppe Maria Barbagallo, Antonio Basile, Pietro Valerio Foti, Stefano Palmucci, Stefano Pergolizzi, Antonio Pontoriero, Corrado Spatola

**Affiliations:** 1Department Scienze Biomediche, Odontoiatriche e delle Immagini Morfologiche e Funzionali, Università di Messina, 98100 Messina, Italy; madalina.larocca@gmail.com (M.L.R.); stefano.pergolizzi@unime.it (S.P.); apontoriero@unime.it (A.P.); 2U.O. Radioterapia Oncologica, A.O.U. Policlinico “G. Rodolico—San Marco” Catania, Via Santa Sofia 78, 95123 Catania, Italy; robertomilazz@hotmail.it (R.M.); lucaliardo@hotmail.com (R.L.E.L.); 3U.O.C. Radioterapia A.O.E. Cannizzaro di Catania, 95100 Catania, Italy; gra.acquaviva@libero.it; 4Division of Neurosurgery, Department of Neurosciences, Policlinico “G. Rodolico—San Marco”, University Hospital, 95121 Catania, Italy; roberto.altieri.87@gmail.com (R.A.); cicciocerto@yahoo.it (F.C.); gbarbagallo@unict.it (G.M.B.); 5Multidisciplinary Research Center on Brain Tumors Diagnosis and Treatment, University of Catania, 95123 Catania, Italy; 6U.O. Radiologia I, A.O.U. Policlinico “G. Rodolico—San Marco”, Via Santa Sofia 78, 95123 Catania, Italy; basile.antonello73@gmail.com (A.B.); pietrofoti@hotmail.com (P.V.F.); spalmucci@unict.it (S.P.); 7Department Scienze Mediche, Chirurgiche e Tecnologie Avanzate “G.F. Ingrassia”, Università di Catania, 95125 Catania, Italy

**Keywords:** glioblastoma, peritumoral zone, reirradiation, stereotactic radiosurgery, proton therapy, altered fractionations, charged particle therapy, radiosensitizers, radiomics, PET/MRI imaging

## Abstract

Glioblastoma (GBM) is the most common and aggressive brain tumor in adults, with a median survival of about 15 months. After the prior treatment, GBM tends to relapse within the high dose radiation field, defined as the peritumoral brain zone (PTZ), needing a second treatment. In the present review, the primary role of ionizing radiation in recurrent GBM is discussed, and the current literature knowledge about the different radiation modalities, doses and fractionation options at our disposal is summarized. Therefore, the focus is on the necessity of tailoring the treatment approach to every single patient and using radiomics and PET/MRI imaging to have a relatively good outcome and avoid severe toxicity. The use of charged particle therapy and radiosensitizers to overcome GBM radioresistance is considered, even if further studies are necessary to evaluate the effectiveness in the setting of reirradiation.

## 1. Introduction

Glioblastoma (GBM) is the most common brain tumor in adults; it represents 54% of all gliomas and 16% of all primary brain tumors. Moreover, it is the most aggressive glioma; in fact, despite the development of a multidisciplinary approach, the outcome for the last four decades has remained stable, with a median survival of about 15 months.

The current standard of care for newly diagnosed glioblastoma is based on cytoreductive surgery, followed by concurrent adjuvant radiotherapy and chemotherapy (CCRT) [1,2,3].

Combined therapy consists of total dose radiotherapy of 60 Gy, delivered in 2 Gy per fraction, over a period of 6 weeks daily along with TMZ 75 mg/m^2^, followed by a maintenance phase with TMZ 150–200 mg/m^2^, 5 days a week every 28 days for six cycles [4,5,6].

Despite the use of all just mentioned multidisciplinary treatments, after a relatively short period of time, almost all GBMs relapse within the high dose radiation field, defined as peritumoral brain zone (PTZ), requiring re-surgery, re-irradiation and or systemic therapies, even if given with palliative intent.

PTZ is defined as a specific parenchyma region containing tumor and stromal cells that promote GBM growth and invasion; moreover, it is characterized by an intratumoral heterogeneity that may influence tumor aggressiveness or response to chemotherapies such as temozolomide. A study published in 1989 pointed out the tendency of glioblastoma multiforme to recur near the initial pre-surgical tumor bed: in detail, it showed that 28% of unifocal tumors recur within 2.0 cm of the pre-surgical initial lesion margin, 56% of tumors recurs within 1.0 cm of the initial lesion margin. This recurrence tendency carries important implications in the planning of external beam irradiation, most importantly supporting the rationale for post-operative treatment with partial brain irradiation rather than whole-brain irradiation [7]. In the present review, the primary role of ionizing radiation in recurrent GBM is discussed and, since therapeutic decision must be individualized, the current literature knowledge about the different RT modalities, dose and fractionation options at our disposal is summarized in order to tailor the treatment to every single patient, balancing the benefits with the risk of toxicity.

## 2. Non-RT Therapeutic Options for Recurrent GBM

For patients affected by recurrent GBM, no standard of care exists, and the median overall survival is less than a year.

Even if in a large cohort study, re-operation demonstrated to not affect survival, for patients in good performance status, with tumor recurrence in non-eloquent areas, a second surgery is currently considered a feasible local approach [8]. In particular, a review published in 2021 strongly suggests re-operation, based on data that total or subtotal resection (extent > 80%) is associated with longer survival [9].

Regarding systemic therapies, even if it is beyond the scope of this review, a brief summary of the evolution and the current state of scientific knowledge is treated to provide context to re-irradiation decision making.

Historically, before temozolomide introduction, the standard treatment for recurrent high-grade gliomas was nitrosourea-based chemotherapy. Perry et al. then conducted a multicenter phase II study to assess the efficacy and safety of continuous dose-intense TMZ for recurrent GBM, concluding that rechallenge with continuous dose-intense TMZ 50 mg/m^2^/d can be considered as a valuable therapeutic option for patients with recurrent GBM, even if it merely provided a 6-month progression-free survival of 24% [10]. In summary, at the current time, TMZ is preferred in the first place because it is well-tolerated, it has good oral bioavailability, and it is convenient to administer as an outpatient regimen. Secondly, nitrosourea-based therapy can be considered as a reasonable alternative, even if the median progression-free survival is usually only 2–3 months and is associated with a 50% grade 3+ hematological toxicities. At the current time, lomustine is used as a second course of alkylating agent to treat most recurrent GBM patients who are eligible for salvage therapy.

After the introduction of TMZ in 2005, despite the development of many preclinical studies and clinical trials, only two new agents have been approved by U.S. Food and Drug Administration: bevacizumab (Avastin), initially approved through the accelerated approval program, and Tumor-Treating Fields (TTFs). In detail, bevacizumab is a humanized monoclonal VEGF-targeting antibody that was shown to improve patient outcomes in combination with chemotherapy (most commonly irinotecan) in recurrent GBM. Moreover, since vascular damage is followed by vascular endothelial growth factor (VEGF) expression at high levels, inhibiting VEGF and acting on vascular tissue around the brain necrosis area, bevacizumab can alleviate brain edema symptoms caused by radiation brain necrosis. Currently, in Italy, this agent has still not been approved, but it is used in Canada, the United States and many other countries outside the European Union for recurrent glioblastoma that has progressed following prior therapy, while in patients with rapidly progressing disease, a combination of bevacizumab and lomustine can be considered [11].

Regarding TTFs, in 2015, a randomized clinical trial investigated their potential as a promising cancer treatment. TTFs therapy involves the continuous delivery of low-intensity electric fields alternating at an intermediate frequency, and this is applicated to the shaved head through a transducer connected to a portable device. This modality was shown to improve both progression-free and overall survival in GBM, and even though it was firstly approved for the treatment of recurrent GBM, this device was subsequently approved by U.S. Food and Drug Administration (FDA) and European Medicines Agency (EMA) even as adjuvant therapy for newly diagnosed supratentorial GBM (due to low cost/benefit ratio, it has not been approved by most European countries) [12].

Another agent that aroused researchers’ interest is regorafenib, an oral multikinase inhibitor of angiogenic, stromal and oncogenic receptor tyrosine kinases, currently approved by the Italian Medicines Agency (AIFA) for recurrent GBM. In 2009, Lombardi et al. published a randomized phase II trial (REGOMA) comparing regorafenib to lomustine in the treatment of recurrent glioblastoma. This trial showed an encouraging overall survival benefit concluding that this drug might be a new potential treatment for these patients; however, further investigation and adequately powered phase 3 study are needed. Regarding other molecular targeted therapies, both clinically and pre-clinically trials were conducted, but currently, these regimens have still not impacted the disease course [13].

## 3. Re-Irradiation

Even though the damage of normal brain tissue previously RT-treated is a reason for concern, re-irradiation currently represents a feasible local approach to use as an alternative or in addition to surgery. In detail, the available evidence, mostly level III, suggests that re-irradiation provides encouraging disease control and survival rates, and in some cases (for example, radiosurgery vs. open surgery at recurrence), is related to a significantly reduced risk of death.

In order to have a relatively good outcome and avoid severe toxicity, it is strongly recommended in the daily clinical routine to tailor the treatment to every single patient in terms of fractionation, dose and constraints.

### 3.1. Stereotactic Techniques, Altered Fractionation and Brachytherapy Treatments

The modalities mostly used are conventionally fractionated stereotactic radiotherapy, hypofractionated stereotactic radiation therapy, stereotactic radiosurgery or brachytherapy techniques, characterized by the different total number of fractions and different total dose per fraction (Figure 1).

At the time of recurrence, stereotactic radiosurgery (SRS) can be considered a valid alternative rather than open surgery in selected patients with small tumor volume (<10 mL), identified by contrast-enhanced magnetic resonance imaging (MRI) sequences. Even though it has a narrower therapeutic window between effective tumor control and radiation necrosis, many studies suggest that SRS is effective mainly for GBMs, more so than other gliomas. In detail, in a prospective study published in 2008, SRS was shown to prolong survival (23 months vs. 12 months; *p* < 0.0001) and to be a safe and effective salvage treatment with acceptable morbidity for selected patients with recurrent small-sized GBMs [14]. Regarding gamma knife surgery (GKS), the first retrospective study that compared surgery and GKS for GBM recurrences in terms of survival outcomes was published in 2012, showing a significantly lower complication rate for GKS and a possible survival benefit for small GBMs, since the median time from the second intervention to tumor progression was longer after GKS than after resection [15].

Even though the median survival in patients with recurrent malignant gliomas following single-fraction SRS is encouraging, the potential toxicity of single fractional treatment is a reason for concern for larger tumors or tumors located close to eloquent structures such as the optic pathway, basal ganglia, motor or speech area. In these cases, it is suggested to exploit the radiobiological advantages of fractionation to improve the therapeutic ratio.

In this regard, a conventionally fractionated stereotactic technique (CFTR) can be used. It is generally based on fractionation schemes of 1.8 Gy × 33 fx or 2 Gy × 30 fx, and despite the longer treatment course, CFRT is indicated for lesions with larger volumes.

Even though the literature data support this technique for its feasibility and the acceptable side effect rates, in the last two decades, CFRT has been superseded in clinical practice by hypofractionated schedules because of its higher degree of precise patient positioning, accurate dose delivery and reduced treatment duration.

In particular, hypofractionated stereotactic radiotherapy (HFRT) was experienced in different studies using fraction sizes ranging from 3 to 7 Gy and a number of fractions from 5 to 10 in the attempt to reduce treatment time. A systematic review published in 2019 concluded that in terms of efficacy and safety, CFRT and HFRT are similar (even though HFRT might result in better survival outcomes in patients aged >70 years), but HFRT has the advantage the treatment being completed in a shorter time, improving the quality of life in patients with poorer prognosis. Moreover, since almost 90% of local recurrences are located within the irradiation field, suggesting that failure to respond to radiotherapy may be attributable to radioresistance, it is reasonable that dose escalation might have some potential benefits in surmounting local recurrence [16].

In the wake of finding the best fractionation schedule optimization (FSO) to delay GBM recurrence, over the years, different strategies have been investigated. In particular, a study published in 2021 tested the efficacy of a super hyperfractionated approach, causing the increase in treatment duration to one year. The variable time intervals between dose fractions used in this study corresponded to the total number of fractions equivalent to weekly, bi-weekly and monthly deliveries (*n* = 53, 27, 13). The recurrence time points were found to be significantly delayed compared with the recurrence time of conventional fractionation; a further recurrence delay was shown to be obtained with a dose escalation to BEDnormal of 150 Gy. Despite the promising results of this novel treatment delivery method, to date, this technique is not used in clinical practice, and more preclinical and clinical studies are needed to validate its efficacy [17].

For the brachytherapy technique, the median dose generally used is 60 Gy (range 40–70 Gy), the median prescribed depth is 5 mm (IQR: 5–7.5 mm), and the radionuclides majorly exploited are I-125 and Ir-192. A systematic review published in 2018 reported a higher OS-12 for brachytherapy when compared to external beam radiotherapy (EBRT), but it is possible that this rate could be due to patient selection bias since patients undergoing brachytherapy are generally surgical candidates with better performance status. Nevertheless, this technique is not commonly used in the modern era, and its role is diminishing as technical experience with the evolving of conformational RT techniques [18].

### 3.2. Safety and Tolerance of Reirradiation

One of the most important key issues for radio-oncologists is the accurate delineation of target volumes and organs at risk (OARs) to precisely calculate the spatial dose distribution and to choose the optimal radiation dose fractionation schedule. In fact, an important component of the decision-making process for second radiation treatment is the expected toxicity induced by the radiation dose delivered to OARs, for example, brain parenchyma, brainstem and optic pathways.

Magnetic resonance imaging and in detail contrast-enhanced T1 and T2 weighted sequences are routinely used to define target volumes. The lesion visible on contrast-enhanced T1 weighted images is defined as gross tumor volume (GTV), while the clinical target volume (CTV) represents the areas of potential suspected microscopic tumor infiltration and potential paths of microscopic spread. In order to define planning target volume (PTV) instead, a further millimetric margin (5 mm or 10 mm) is added. Since radiosurgery does not require correcting the set-up errors, in all the series where patients are treated with a gamma knife, no margin for PTV is generally used.

For brain parenchyma radiation-induced toxicity, the major complication is radionecrosis, which can be radiologically diagnosed, histologically proven or associated with symptoms. Since the relapse typically occurs in-field or marginal to the field of the first-course treatment (peritumoral zone), the risk of severe side effects increases due to the overlapping of the target volume with the volume previously treated.

In 2008, Mayer et al. published a review about radiation tolerance of the human brain, concluding that the normal brain tissue necrosis is related to the cumulative equivalent dose in 2 Gy (EQD2) per fraction: >100 Gy for patients treated with conventional external beam RT, >105 for fractionated stereotactic radiotherapy and >135 for radiosurgery. In other words, patients with GBM who had received prior radiotherapy with a total dose of 60 Gy in 2 Gy fractions could receive reirradiation for an additional EQD2 of 40 Gy, 45 Gy and 75 Gy with conventional external beam radiotherapy, fractionated stereotactic radiotherapy and radiosurgery, respectively. The authors commented that the applied reirradiation dose and EQD2 cumulative were found to increase with a change in the irradiation technique from a conventional to a conformal technique, such as FSRT to radiosurgery retreatment, without increasing the risk of normal brain necrosis [19].

A review published in 2018 showed that the volume of the target influences the risk of toxicity and, consequently, the choice of fractionation: radiosurgery with EQD2 < 65 Gy may be a choice for small lesions (target volume < 12.5 mL, Figure 2), hypofractionated stereotactic radiotherapy with EQD2 < 50 Gy is feasible for medium lesions (target volume up to 35 mL), whereas conventionally fractionated treatment with EQD2 < 36 Gy may be used for reirradiation of large lesions (target volume up to 50 mL).

For the brainstem and the optic pathways, it is necessary to meet the classical constraints and tolerance dose reported by the QUANTEC review, even if it is likely that they had already received doses near to the constraints during the first radiation course. For the entire brainstem, the tolerated dose is 54 Gy in 1.8/2 Gy fractions, while a volume of 1–10 mL can tolerate a total dose of 59 Gy in 1.8/2 Gy. For radiation-induced optic neuropathy, the incidence is 3–7% for doses of 55–60 Gy, and it increases for doses >60 Gy up to 7–20% [20,21].

Regarding stereotactic radiosurgery-related toxicities, the most reported is radionecrosis, histologically or radiologically proven; other severe neurologic toxicities observed are acute herniation, hydrocephalus and cranial nerve deficit. Scoccianti et al. reported a rate of severe toxicity <3.5% in all the series with a median target volume <12.5 mL and a median prescription dose <15 Gy, suggesting that patients with small target volume may be eligible for SRS and that median prescription dose should remain between 12 and 15 Gy [20].

For conventionally fractionated radiotherapy, a dose of 36 Gy in 2 Gy fractions seems to be safe for volumes up to 50 mL, while for hypofractionated stereotactic radiation therapy, the toxicities are highly variable and comprehend radionecrosis, hydrocephalus, dizziness or other neurological symptoms. Such as for SRS, the tolerability of HFSRT was related to the extent of target volume: despite the choice of a schedule with low EQD2 (43.75 Gy), Cho et al. reported an 8% of severe toxicity rate for a median target volume of 40 mL, while Kim et al. reported a very high rate of histologically confirmed radionecrosis. The literature analysis suggests that HF regimes with an EQD2 < 50 Gy have an optimal toxicity profile (severe toxicity rate < 3.5%) when used in lesions <35 mL [20,22,23].

For brachytherapy, severe RT-related toxicities (Grade 3+) are poorly reported, with half of the studies not reporting toxicity rates. Mainly, radiation necrosis and grade 3–4 hematological toxicities are reported.

### 3.3. Charged Particle Therapy

Recent developments in radiotherapy technology have led to the availability of new beam qualities: radiotherapy with carbon ions (CIRT) or protons (PBRT). The interest in these techniques is related to their peculiar dosimetric advantage, compared to photon beam radiotherapy, and the possible ability to overcome GBM radioresistance, influencing OS and PFS rates (Table 1).

In detail, carbon ions characteristics are the inverted dose profile and high local dose deposition within the Bragg peak, which lead to a precise dose application and consequently to spare as much normal tissue as possible. Moreover, in comparison to photons, carbon ions offer an increased relative biological effectiveness (RBE), which can be calculated between 2 and 5.

The first study that evaluated carbon ion radiotherapy for recurrent gliomas is the CINDERELLA trial, published in 2010, which tried to define the optimal recommended dose of carbon ion radiotherapy for re-irradiation, and then demonstrate the superiority in survival compared to FSRT. Then, a study published in 2021 investigated overall survival in recurrent glioblastoma treated with either carbon ion reirradiation or photon reirradiation. This study demonstrated CIRT to be a feasible treatment option for recurrent glioblastoma mainly thanks to its better dosimetry which allows safe irradiation of a large median PTV with high doses (45 Gy carbon ions, 15 fractions, EQD2 = 48.8 Gy, α/β = 10 Gy for tumoral tissue; EQD2 = 56.3 Gy, α/β = 2 Gy for OARs) sparing, at the same time, the surrounding organs at risks [24,25,26].

For proton beams radiotherapy (PBRT), its dosimetry advantage is related to the specificities of linear energy transfer, called Bragg peak: in particular, this technique is characterized by a low incoming dose and the majority energy deposition to a single point, usually, the target, beyond which the energy declines sharply. Since GBM generally relapse within the high dose radiation field (PTZ), the Bragg peak, together with the possibility to not use coplanar beams, including vertex beams, allow a minimized reirradiation of previously irradiated brain tissue and a spared cumulative radiation dose to critical OARs, allowing to increase the total dose [27,28].

In 2020, Saeed et al. analyzed treatment patterns, toxicities and clinical outcomes of patients who underwent PBT reirradiation, collecting the largest analyses about PBT reirradiation of patients with recurrent GBM to date. In this study, it was confirmed that PBRT is well tolerated and offers efficacy rates comparable with previously reported photon reirradiation. In the same year, Scartoni et al. affirmed that PBRT is also able to preserve the health-related quality of life (HRQOL) until the time of disease progression in addition to being a safe and effective treatment [29,30,31,32].

## 4. Predictive Factor of Response to Re-Irradiation and Combination with Systemic Therapies

For years, a clinical practice documented that not all patients respond to adjuvant therapies in a similar way.

Specifically, some patients are very susceptible to recurrence due to insensitivity to adjuvant therapy, while other patients demonstrated to be more sensitive, recording significantly longer progression-free survival and overall survival. This heterogeneity in therapeutic response seems to be related to both cell-intrinsic molecular factors, such as genetic variations, and micro-environmental factors.

### 4.1. Role of IDH1 and MGMT

Regarding genetic variations of GBM, in 2008, Parsons et al. published an integrated genomic analysis of human glioblastoma multiforme revealing an Isocitrate Dehydrogenase 1 mutation (IDH1), located at codon R132, which is characteristic of all subtypes of glioma, excepting for primary GBM [33]. Evidence suggested that GBM with IDH1 mutation represents a distinct disease entity characterized by a different clinical behavior: it emerges in the frontal lobe in young individuals and has a significantly longer PSF and OS [34].

The mechanism behind the improved survival in the subset of a patient with IDH1 mutation seems to be related to cells’ increased susceptibility to oxidative damage and consequently to the increased sensitivity to radiotherapy (but not to temozolomide) [35,36,37].

Currently, three Isocitrate Dehydrogenase enzymes have been identified (IDH1, IDH2 and IDH3), but only IDH1 and IDH2 enzymes were shown to be mutated in GBM.

Other than IDH mutations, during past decades, another prognostic factor of equal importance has been identified: the O6-methylguanine DNA methyltransferase (MGMT) promoter methylation, a genomic modification able to define patients who are most likely to benefit from the addition of temozolomide chemotherapy and to foresee the recurrence pattern (in the field (PTZ) or distant to the primary radiation field). Although the factors underlying this phenomenon are not clear, it is possible that MGMT methylation status can alter the motility and migration pattern of GBM cells and that the combined chemoradiation approach can enhance local GBM cell eradication thanks to the synergic effect on MGMT methylated cells. These hypotheses are supported by a study published in 2009, which demonstrated that the time to recurrence was prolonged in patients with MGMT methylated status, as was the time to distant recurrence. Furthermore, the recurrence pattern was significantly influenced by MGMT methylation status: recurrences happened outside the RT field in 15% and 42% of patients with MGMT unmethylated and MGMT methylated status, respectively [38,39]. Moreover, a recent study suggested that dual alkylator therapy with temozolomide and lomustine might improve survival compared with standard temozolomide therapy in patients with newly diagnosed glioblastoma with an MGMT-methylated promoter [40].

### 4.2. Radiosensitizers

Radiosensitivity is a parameter that reflects the susceptibility of cells to succumb to ionizing radiations (IR), which perform their cytotoxic effects by inducing DNA double-strand breaks (DSB) and lethal chromosomal aberrations.

Since GBM has shown to be highly resistant to IR, a review published in 2020 tried to identify all the micro-environmental factors that may be involved in this process and the possible benefit of radiosensitizers [41].

In many clinical trials, some established chemotherapeutic agents have been used as radiosensitizers and were reported to enhance the efficacy of radiotherapy successfully: in 2009, Sigmond et al. demonstrated for the first time that gemcitabine (GEM) is able to pass the blood-tumor barrier in GBM patients; moreover, in this study, both gemcitabine and dFdU concentrations in tumor samples were high enough to enable radio-sensitization. A year later, a phase II study was conducted in order to evaluate the activity of gemcitabine as a radiosensitizer for newly diagnosed GBM; this trial showed that concomitant radiotherapy-gemcitabine is actually active both in tumors with methylated and unmethylated MGMT promoter and it is well tolerated [42,43].

Currently, even if GEM is widely used for the treatment of various solid tumors as a single agent or in combination with other chemotherapeutic drugs, its use against GBM has only been evaluated in preclinical and clinical trials, and further investigations are needed.

For the radio-sensitization role of TMZ, Palanichamy et al. published a review that identified TMZ’s role in radiation-induced DNA damage stabilization when administered with radiation. In detail, through activation of the mismatch repair (MMR) and ataxia telangiectasia-mutated (ATM) pathways, TMZ was shown to induce cells arrest in the more susceptible G2/M phase and to enhance the DNA-damaging effects of radiation [44]. Currently, TMZ is widely used both in the post-operative setting (CCRT) and as maintenance.

In 2014, Setua et al. published a study reporting the first successful chemo-radiotherapy on patient-resistant GBM cells using a cisplatin-tethered gold nanosphere. The theoretical bases for this study were that after intracellular uptake, the nanosphere could promote DNA damage leading to caspase-mediated apoptosis. In the presence of radiation, both gold and platinum of cisplatin serve as high atomic number radiosensitizers leading to the emission of ionizing photoelectrons and Auger electrons. This resulted in enhanced synergy between cisplatin and radiotherapy mediated cytotoxicity and photo/Auger electron mediated radio-sensitization, leading to complete ablation of the tumor cells in an in vitro model system. Even if not used in clinical practice, this study demonstrates the potential of designed nanoparticles to target aggressive cancers in patient-derived cell lines, providing a platform to move towards treatment strategies [45].

In recent trials, other agents that were evaluated as radiosensitizers are PARP (Poly-ADP-Ribose-DNA Polymerase) inhibitors, a family of proteins implicated in the base excision repair (BER) pathway. PARADIGM and PARADIGM-2 trials evaluated the combination of olaparib with RT demonstrating that radiation with olaparib is actually well-tolerated. While in the VERTU trial, veliparib with chemoradiation and TMZ combination was evaluated in MGMT-unmethylated GBM, demonstrating that this association is also well tolerated but did not improve outcomes [46,47]. None of these agents are currently used in clinical practice.

In conclusion, considering that radiotherapy represents an integral component of the standard of care therapy for GBM, the use of radiosensitizers could have a crucial impact on the disease course. For this reason, radiosensitizers have been widely considered and currently remain a viable option for improving the outcome of therapy in GBM, even if they have not yet achieved this potential. Overall, more research is necessary to fully understand the mechanisms of GBM radioresistance and improve the outcomes of patients with this deadly disease.

## 5. The Role of Radiomics and PET/MRI Advanced Imaging in the Management of Relapsing GBM

Reirradiation of high-grade gliomas represents a difficult challenge for the radiation oncologist because GBMs tend to relapse within the high-dose radiation field, defined as the peritumoral brain zone. Therefore it is necessary to re-irradiate an already treated issue (approximately 4–6 months before) to prevent further neurological, motor and cognitive toxicity.

In detail, PTZ is a specific parenchyma region containing tumor and stromal cells able to promote GBM growth and invasion. This zone has been studied for years to identify PTZ characteristics and to define the best therapeutic approach. It was first shown that tumor cell infiltration could be detected in areas considered normal both on standard MRI and by the neurosurgeon under an operating microscope. Secondly, it is provided with selected tumor clones and stromal cells with tumorigenic and angiogenic properties that increase GBM aggressivity [48,49,50].

Furthermore, in recent years, there has been a rapid implementation of radiotherapy techniques with the use of intensity-modulated radiotherapy (IMRT), volumetric modulated arc therapy (VMAT) and stereotaxic brain treatments (SBRT), which allow delivering a high dose of radiation to smaller volumes with more precision, while sparing the surrounding healthy tissues. In parallel, we assisted the implementation of advanced radiodiagnoses, such as radiomics, MRI and PET images, which demonstrated to be fundamentals to achieve better precision in the radiation treatment of the primitive and relapse [51,52].

### 5.1. Radiomics

Radiomics is a process that consists in the transformation of PET/CT/MRI images to mineable data; it implies image acquisition, segmentation and labeling of the tumor/normal tissues, extraction of quantitative features (shape, texture, intensity), followed by machine learning and statistical modeling. This technology can be used in the pre-surgical, radiation or post-treatment phase; moreover, it may play a role both in the staging and grading phase. In 2021, Russo et al. exploited machine learning on 11[C]-MET PET/CT scan images of fifty-six patients affected by a primary brain tumor to create a predictive model capable of discriminating low- and high-grade CNS, demonstrating a percentage of sensitivity of 72–86.7%, and specificity and accuracy greater than 80% [53].

Radiomics also play a role in differentiating post-treatment effects such as pseudoprogression (PsP) from true progression worthy of re-RT, which is one of the major problems for the multidisciplinary neuro-oncology team. In detail, PsP is defined as a transient magnetic resonance imaging pattern that use to mimic tumor progression. It generally occurs during the first 3 months after radiation therapy and improves within a few weeks or months; this phenomenon is more frequent in patients treated with concomitant temozolomide than in those receiving radiation therapy alone, and it is particularly frequent in patients MGMT promoter methylation. In this regard, current studies support the potential clinical applications of radiomics to predict pseudoprogression. For example, in 2021, Baine et al. made published a retrospective review that analyzed radiomic data of the pre-RT MR images of 72 patients, mostly treated with 60 Gy in 30 fractions with concomitant temozolomide: of these patients identified for the study, 35 were able to be assessed for pseudoprogression, and 8 (22.9%) had pseudoprogression [54].

### 5.2. The Use of PET/MRI

PET/MRI images are also very useful in the re-RT planning phase of malignant CNS tumors.

Many authors attempted to compare MRI and PET in CT fusions for re-RT of relapsed HGGs. One of the first comparisons between the use of PET and MRI in the re-RT of relapsed HGG was performed by Grosu et al. In this study, 44 patients with relapsed HGGs were re-treated with a total dose of stereotaxic treatment (SRT) of 30 Gy in six fractions. The gross tumor volume (GTV) was defined by CT/T1 gadolinium-MRI image fusion in 18% of the patients and by 11C-methionine positron emission tomography (MET-PET) or 123I-alpha -methyltyrosine (IMT) single-photon computed emission tomography (SPECT)/computed tomography (CT)/magnetic resonance imaging (MRI) fusion in 82% of the patients, 66% of patients associated chemotherapy with temozolomide. In this study, the median survival time was 9 months for treatment planning based on PET(SPECT)/CT/MRI vs. 5 months for treatment planning using only CT/MRI. Median survival times were 11 months for patients who received SRT based on biologic imaging plus temozolomide and 6 months for patients treated with SRT without biologic imaging, without temozolomide or without both [55,56].

In 2016, Oehlke et al. treated 44 patients with relapsed HHG with re-RT. These patients were previously treated with 59.4–60 Gy (single dose 1.8–2.0 Gy) and were all patients ineligible for surgery or with macroscopic residue. The size of the relapsed tumor ranged from 1 to 6 cm. They randomized the patients into two arms: in the first arm, the GTV was identified on MR images, in the second one used FET-PET images. In both arms, CTV is defined by adding 3 mm in either direction respecting anatomical boundaries such as the skull and/or tentorium. This CTV was then expanded to the PTV by adding 1–2 mm in all directions. The dose of stereotaxic re-RT performed was 39 Gy, 3 Gy/day, 5×/week to the PTV. The aim of the study was to compare PFS and OS in the two arms and the site of relapse. Considering that PET and MR recurrence images do not always overlap, PTV can vary greatly. It will be interesting to have the future results of this phase III study [57].

In 2014, Miwa et al. treated 22 patients with relapsed HGG with hypo-fractionated SRT planned with 11C-methionine positron emission tomography (MET-PET)/computed tomography (CT)/magnetic resonance imaging (MRI) fusion (Figure 3). PTV was obtained from GTV, in detail from PET/MR images plus 3 mm margin. The total delivered dose was 25–35 Gy with fractions of 5–7 Gy, respectively. The median overall survival time was 11 months, and the median progression-free survival time was 6 months from the date of re-irradiation, respectively. The OS of patients who received combined TMZ chemotherapy was greater. The authors proved the usefulness and importance of PET in the planning and follow-up of re-RT. They compared MRI and MET-PET images of the relapse and noted that only a part of the lesion that can be seen in MRI had MET high-uptake. The GTV was identified precisely on the MET uptake alone. Five months after the SRT, an MRI was performed, which indicates an increase in the size of the irradiated lesion. The control PET, however, does not show any area of MET high-uptake and is therefore suggestive of pseudoprogression [58].

Fleischmann et al. subjected seven relapsed HGG patients to radiation retreatment. The fractionation was 36 Gy in fractions of 2 Gy/day. They used the PTV obtained from the 18F-FET PET by blending it with the GTV obtained from the MRI and obtained the PET-MRGTV. They added 10 mm to this volume to obtain the PTV on planning CT. Six patients were treated with concomitant bevacizumab, and one was treated with temozolomide. They later created PET-MRGTV with margins of 10, 8, 5 and 3 mm. They performed clinical-instrumental follow-up with MRI for five patients, two were lost and one patient developed distant re-relapse; four patients instead developed re-relapse within the PET-MRGTV with a 3 mm margin. The median volume of the PET-MRGTV with a margin of 3 mm was 91.3 cc and, therefore, almost half the size of the median original PTV volume of 177.7 cc. This study, albeit with its numerical limits, offers important reflections on the volumes to be re-radiated, considering the previous radiant treatment. If this data were confirmed on a large scale, a higher dose could be delivered, considering that 60 Gy at primary treatment and 36 Gy at re-RT are insufficient for long-term disease control [59].

The use of PET and MR in planning the re-RT of relapsed HGGs could also become strongly recommended for radiotherapy with carbon ions. In this regard, Debus et al. retrospectively investigated the importance of PET images in relapsed HGG patients treated with carbon ions. The retrospective study included 26 patients treated with carbon ion radiotherapy with a total dose of 30–42 Gy in fractions of 3 Gy. All patients underwent MR and O-(2-[18F] fluoroethyl)-l-tyrosine (18F-FET) PET before treatment, but only MR and planning CT were used for GTV. GTV enclosed the contrast-enhancing structures on T1 MRI images and, if applicable, resection cavities from antecedent treatments. With a few exceptions, they saw that most of the PET volumes (~90%) are included in the T2 MRI FLAIR hyperintense volumes (CTVFLAIR) in both grade III and IV glioma. They also found that about 83% of re-relapses are in the marginal zone of CTV in patients treated with carbon ions, as opposed to patients treated with conventional radiotherapy who relapse in the field for >90% [60].

## 6. Conclusions

Even though the damage of normal brain tissue previously RT-treated is the reason for concern, re-irradiation currently represents a feasible local approach to use as an alternative or in addition to surgery. In order to have a relatively good outcome and avoid severe toxicity, it is strongly recommended to use radiomics, MR and PET images to achieve better precision and to tailor the treatment to every single patient choosing different fractionations, according to different target volumes, and when available, high conformality techniques to overcome GBM radioresistance.

## Figures and Tables

**Figure 1 brainsci-12-00416-f001:**
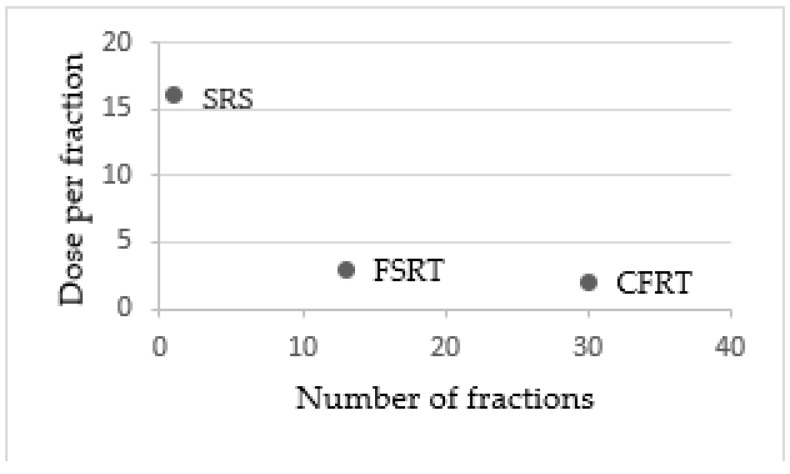
Differences in terms of fractionations between stereotactic radiosurgery (SRS), fractionated stereotactic radiotherapy (FSRT) and conventionally fractionated radiotherapy (CFRT).

**Figure 2 brainsci-12-00416-f002:**
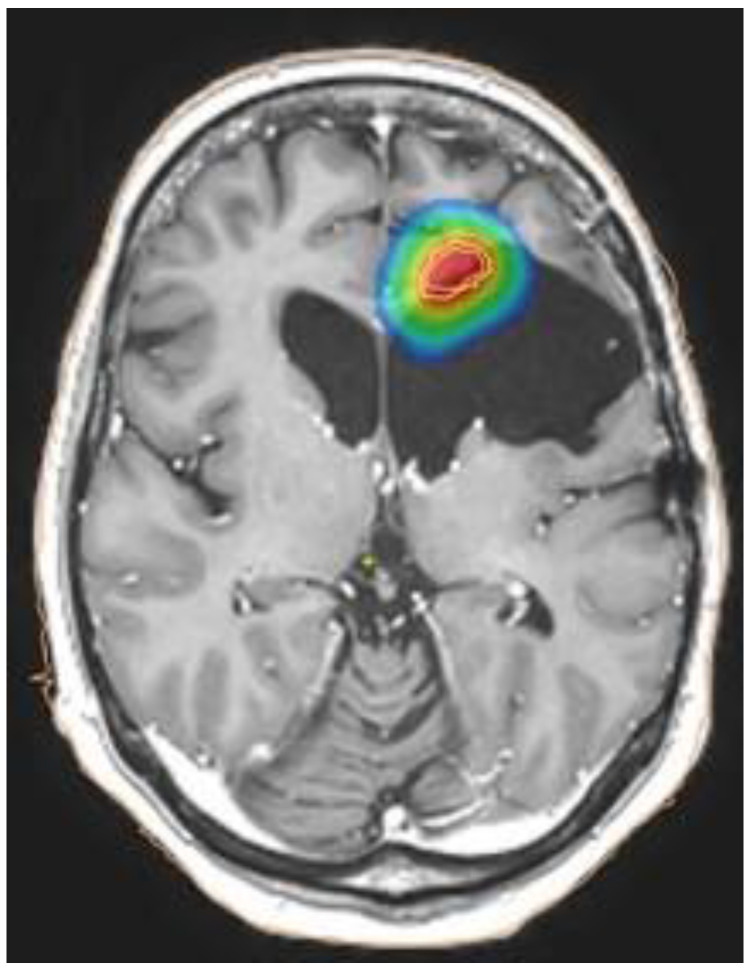
Example of dose distribution for small lesion treated with stereotactic radiotherapy: the colors correspond to an EQD2a/b = 2 of blue > 8 Gy; green > 30 Gy; yellow > 40 Gy; red > 50 Gy.

**Figure 3 brainsci-12-00416-f003:**
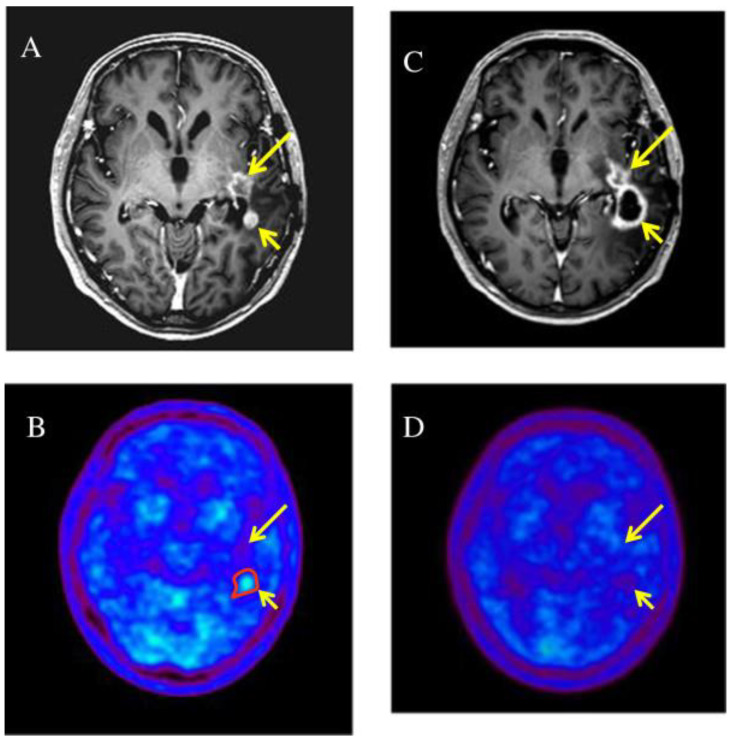
Two enhanced lesions (long and short arrow) were demonstrated in the left temporal lobe on T1-weighted magnetic resonance imaging (**A**), MET-PET demonstrated a MET high-uptake on the region of short arrow (**B**), only the enhanced lesion (short arrow) was treated with RT; 5 months later it was increased in size (**C**) but not in uptake (**D**) (suggestive of pseudoprogression) while the non treated lesion remained stable [58].

**Table 1 brainsci-12-00416-t001:** Differences in terms of overall survival (OS) and progression-free survival (PFS) between carbon ions radiotherapy, photon beams radiotherapy and proton beam radiotherapy.

Authors	Technique	Schedules	OS	PFS
Lautenschlaeger F.S. et al. [24]	Carbon Ionsradiotherapy (CIRT)	45 Gy, median fraction size 3 Gy per fx	8.0 months	5.5 months
	Fractionated stereotactic radiotherapy with photons (FSRT)	39 Gy, median fraction size 3 Gy per fx	6.5 months	3.9 months
Saeed A.M. et al. [25]	Proton beam therapy (PBRT)	46.2 Gy (range, 25–60 Gy), median fraction size 2.2 per fx	14.2 months	13.9 months

## Data Availability

All the supporting data are presented in the manuscript.

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
