# Peer review of "Relapsing High—Grade Glioma from Peritumoral Zone: Critical Review of Radiotherapy Treatment Options"

_brainsci, 2022, doi:10.3390/brainsci12040416_

Round 1

Reviewer 1 Report

The authors have written a review about re-irradiation for glioblastoma recurrence in the peri-tumoral zone. The topic is very important, as treatment options for recurrent glioblastoma are somewhat limited. The work is intended to a be a critical analysis (based on the title) but most of the exposition just summarizes the literature with significant critical analysis. The English language use is problematic and make the manuscript somewhat difficult to follow, as well. A number of changes would augment the manuscript, as follows:

  1. In section 3.1, the authors should define "small" GBM - as in line 145.
  2. In section 3.1, the authors should indicate how often these re-irradiation techniques are being used, as well as how often they could be used.
  3.  In section 3.1, the authors should indicate where brachytherapy is currently being used. Most U.S centers have not used the technique in decades, but perhaps European centers still use it. Or is its mention just for completeness.
  4. In section 3.1, the authors should clearly define the radiosurgey target. If they mean the contrast-enhancing lesion, they should clearly state that. 
  5. In section 3.2, a reference(s) is needed for lines 262-264.
  6. In section 4.1, the authors need to link their discussion of IDH and MGMT to re-irradiation. Otherwise, the section should be deleted, as it is off topic. 
  7. In section 4.2, the authors should discuss how often radiosensitizers are being used and how often they could be used.
  8. In section 5.1, an example of radiomics use would be helpful.
  9. In section 6, the CONCLUSIONS is too long. Three of four sentences should be sufficient.
  10. In FIGURE 2, the dosimetry dose should be labeled.
  11. In FIGURE 3, the authors need to provide more explanation of what the figure is actually showing. What do arrows mean, for instance? 

Author Response

  1. In line 155 I specified that “small” means a tumor volume < 10 mL (reference n. 14).
  2. As I reported in section 3.1: SRS represents an alternative to surgery only for selected patients with a tumor volume < 10 mL, CFRT has been widely used in the last decades but it has been superseded in clinical practice by hypofractionated schedules, super hyperfractionated RT is a novel treatment recently investigated but further studies are needed to validate its efficacy and it is not currently used in clinical practice.

  3. Brachytherapy is mentioned just for completeness, as I wrote in lines 209-211: “this technique is not commonly used in the modern era and his role is diminishing as technical experience with the evolving of conformational RT techniques”.

  4. In line 155 I defined the radiosurgery target as a lesion “identified by contrast enhanced Magnetic Resonance Imaging (MRI)”.

  5. In section 3.2 all data are taken from references n. 19-23.

  6. I summed up the section and added some specifications. In particular: IDH is related to GBM increased sensitivity to radiotherapy while MGMT methylation is related to the recurrence pattern, either in field (PTZ) or distant to the primary radiation field.

    I reported a study that showed that recurrences happen outside the RT field in 15% and 42% of patients with MGMT unmethylated and MGMT methylated status, respectively. In this regard, I replaced the reference n. 39.

    I believe that this topic, if correctly focused, can be important to foresee the recurrence patter and consequently the radiotherapy treatment planning.

  7. As I reported in section 4.2: to date, gemcitabine is GEM is widely used for the treatment of various solid tumors but its use against GBM has only been evaluated in preclinical and clinical trials. TMZ is widely use, both for concurrent adjuvant and maintenance therapy. Cisplatin-tethered gold nanosphere such as PARP inhibitors are being investigated but they are not used in the clinical practice.

  8. I provided a better explanation of radiomics use in the lines 472-485.

  9. I summed up conclusions.

  10. I added to lines 252-253: “the colors correspond to an EQD2a/b =2 of blue >8 Gy; green >30 Gy; yellow >40 Gy; red >50 Gy”.

  11. I provided to explain the figure in lines 525-532: “two enhanced lesions (long and short arrow) were demonstrated in the left temporal lobe on T1-weighted magnetic resonance imaging (A), MET-PET demonstrated a MET high-uptake on the region of short arrow (B), only the enhanced lesion (short arrow) has been treated with RT, 5 month later it was increased in size but not in uptake (suggestive of pseudo progression), while the non treated lesion remained stable”.

Reviewer 2 Report

This is an extensive review article detailing literature related to physical treatment options available at time of local relapse of glioblastoma. It predominantly describes the rationale and techniques relating to re-irradiation; and data relating to outcome.

Unfortunately although this is a detailed manuscript utilizing many important references, the structure of the manuscript should be revised to allow improved understanding by the reader. These changes should be essentially minor and may involve removal of unnecessary paragraphs.

  1. The key subject for the literature review is local relapse after glioblastoma (in the peritumoural zone). This is not a commonly applied term and is not well defined in the manuscript. The authors appear to relate this zone to prior RT volumes rather than a neuroanatomical zone. I would recommend the introduction to have a greater emphasis of describing this zone and explore further data relating to patterns of failure in glioblastoma ( and potentially impact of MGMT on this pattern of failure).
  2. The structure of the manuscript should be better described in the introduction, as it is hard to comprehend the topics that are to be discussed in the review. For instance salvage second-line systemic therapies are not discussed, nor is the role of salvage surgery ( both of which are utilised as initial salvage prior to a physical therapy such as ReRT, TTF). Thus this manuscript has an emphasis on repeat radiation therapy interventions, and as such that should be reflected in the introduction.
  3. The detailed description of the Stupp protocol is not required in the introduction, and can be removed to allow more focussed information as above.
  4. The detailed biological description of both IDH and MGMT are not relevant to the topic, and most clinicians who are at a level of experience to contemplate ReRT would be familiar with these aspects.
  5. The concept of radiomics has been overemphasised in importance in the conclusion and this should either be removed or have more data provided in the body of the manuscript to validate the conclusion
  6. There has been no discussion on role of bevacizumab  in regards to minimising the subsequent effects of ReRT, which is important given this is utilised in many ReRt regimens.
  7. More detail could be provided in describing the issues related to pseudoprogression in the peritumoural zone, and the difficulties in distinguishing treatment effect from relapse.

Author Response

  1. I added more informations about PTZ both in section 1 and 4.1, specifying that this zone characteristic ad MGMT methylation status influence the recurrence pattern (and consequently the radiotherapy treatment planning).
  2. I summed up the introduction reducing both salvage surgery and salvage second-line systemic therapies discussion to make it more understandable.

  3. See answer n. 2
  4. I summed up the section and add some specifications. In particular: IDH is related to GBM increased sensitivity to radiotherapy while MGMT methylation is related to the recurrence pattern, either in field (PTZ) or distant to the primary radiation field.

    I reported a study that showed that recurrences happen outside the RT field in 15% and 42% of patients with MGMT unmethylated and MGMT methylated status, respectively. In this regard, I replaced the reference n. 39.

    I believe that this topic, if correctly focused, can be important to foresee the recurrence patter and consequently the radiotherapy treatment planning.

  5. I summed up conclusions section but kept the mention to radiomics just for completeness.

  6. I syntetically explained how bevacizumab acts alleviating brain edema symptoms caused by radiation brain necrosis (lines 104-107).

  7. I added more details about pseudoprogression in lines 475-482.

Round 2

Reviewer 1 Report

The authors have addressed the majority of the reviewer's comments. The first sentence of the CONCLUSIONS is not necessary and can be eliminated. The remainder of the manuscript still requires English language editing.

Author Response

I proceeded to eliminate the sentence.